# Surface Functionalization of Bioactive Hybrid Adsorbents for Enhanced Adsorption of Organic Dyes

**DOI:** 10.3390/ijerph20095750

**Published:** 2023-05-08

**Authors:** Yasser M. Riyad, Taha M. Elmorsi, Mohd Gulfam Alam, Bernd Abel

**Affiliations:** 1Department of Chemistry, Faculty of Science, Islamic University of Madinah, Madinah 42351, Saudi Arabia; gulfam@iu.edu.sa; 2Department of Chemistry, Faculty of Science, Al-Azhar University, Cairo 11884, Egypt; taha_elmorsi@azhar.edu.eg; 3Institute of Chemical Technology, Leipzig University, Linne´-Strasse 3, 04103 Leipzig, Germany; bernd.abel@uni-leipzig.de

**Keywords:** mango seed extract-based bioadsorbent, copper-doped zinc oxide/extract adsorbent, single and binary adsorption, kinetic models, adsorption isotherms

## Abstract

In this study, a valuable adsorbent was functionalized using commercial ZnO and a mango seed extract (MS-Ext) as a green approach for synthesis. Fourier-transform infrared spectroscopy, transmission electron microscopy, scanning electron microscopy, and energy-dispersive X-ray analysis spectraconfirmed the presence of bioactive phenolic compounds and ${\mathrm{Cu}}^{2+}$ ions on the surface of ZnO. Functionalized Cu-doped ZnO/MS-Ext exhibits high efficacy in acidic, neutral, and alkaline medium, as indicated by 98.3% and 93.7% removal of methylene blue (MB) and crystal violet (CV) dyes, respectively. Cu-doped ZnO/MS-Ext has a zeta potential significantly lower than pristine zinc oxide (p-ZnO), which results in enhanced adsorption of cationic MB and CV dyes. In binary systems, both MB and CV were significantly removed in acidic and alkaline media, with 92% and 87% being removed for CV in acidic and alkaline media, respectively. In contrast, the removal efficiency of methyl orange dye (MO) was 16.4%, 6.6% and 11.2% for p-ZnO, ZnO/Ext and Cu-doped ZnO/Ext, respectively. In general, the adsorption kinetics of MB on Cu-doped ZnO/MS-Ext follow this order: linear pseudo-second-order (PSO) > nonlinear pseudo-second-order (PSO) > nonlinear Elovich model > linear Elovich model. The Langmuir isotherm represents the adsorption process and indicates that MB, CV, and MO are chemisorbed onto the surface of the adsorbent at localized active centers of the MS-extract functional groups. In a binary system consisting of MB and CV, the maximum adsorption capacity (q_m_) was 72.49 mg/g and 46.61 mg/g, respectively. The adsorption mechanism is governed by electrostatic attraction and repulsion, coordination bonds, and π–π interactions between cationic and anionic dyes upon Cu-doped ZnO/Ext surfaces.

## 1. Introduction

Ever-changing industrial development accompanied by the discharge of contaminants to surface water bodies raises new environmental challenges posing direct threats to ecological equilibrium, human health, and aquatic life [1]. Wastewater discharged by dye-related industries including cosmetics, food, paper, and textile contains significant amounts of dyes that upon mixing with the water bodies result in high levels of chemical oxygen demand as well as light penetrability constriction [2]. In addition, most of these contaminants are toxic, comprising carcinogenic and mutagenic properties to aquatic organisms [3,4]. Therefore, treating dye-containing wastewater before being discharged into water bodies is becoming more crucial than ever [5]. Numerous treatment methods have been deployed, including (a) physical methods, such as physical sorption, ion exchange, and membrane filtration [6,7,8], (b) chemical methods, such as redox treatment, precipitation, and photocatalysis [9], and (c) biological methods, such as aerobic/anaerobic treatment [5,10]. Among these treatment methods, adsorption is particularly considered one of the most attractive methods due to its high efficiency, simplicity, low cost, low energy consumption, and recyclable nature [6,11,12,13,14] Through extensive research, various materials [15,16] have been studied as dye-removal adsorbents, including activated carbons [17], biochar [18,19], zeolites, alumina, silica gel, graphene oxide [20], natural materials (wood, coal, chitin/chitosan, clay), industrial/agricultural/domestic wastes, and nanomaterials [21,22,23,24]. Adsorption potential of nanomaterials (spherical ZnO, chitosan and ZnO modification) [24,25] for a wide variety of organic and inorganic contaminants is due to their higher surface area and reactivity. However, a lack of selectivity towards specific organic dyes from their mixtureshas been found [12]. Therefore, selective adsorption and separation of organic dyes from complex dye mixtures have receivedgrowing interest.

It is crucial to design materials with a simple preparation process, good separation capability, and high adsorption efficiency of dyes from wastewater [26,27,28,29]. Pristine and surface-functionalized CNCs have been used as dye absorbents [12]. Introducing functional groups containing different charges provided the cellulose nanocrystals with the capability of selectively binding with cationic or anionic dyes. It was reported that about 20 million tons of the waste generated during the processing of *Mangifera indicafruit* (mango) caused a series of environmental and economic problems [30], although mango seeds are used as raw material to obtain phenolic compounds [31,32] (such as phenolic acids, tannins, flavonoids, and xanthones) with biological activities (such as antioxidant, antimicrobial, anticancer, etc.) [32]. It is environmentally significant to develop different methods to use the waste of mango seeds in different applications, which will be a benefit to society [30]. More research is needed on the inclusion of mango phenolic and polyphenol [31] compounds in different materials, such as metal oxide nanoparticles (MONPs), to improve their barrier and physicochemical properties [30]. In this study, we suggested using mango seed kernel powder extract (MS-Ext) for surface functionalization of commercial ZnO powder as a green chemistry approach to synthesis of a bioactive hybrid adsorbent. MS-Ext, a by-product of the processing industry, is a potential source of bioactive phenolic compounds (BPC) with different functional groups, which may alter the surface charge of commercial metal oxides and enhance the adsorption and separation of mixtures of organic dyes. To investigate this hypothesis, we proposed a method for surfacefunctionalization of pristine ZnO (p-ZnO) using mango seed kernel extract (MS-Ext) to obtain different adsorbents of ZnO/MS-Ext. The results obtained with p-ZnO were then compared with those from modified adsorbents based on the adsorption and selective binding of dye pollution models such as methylene blue (MB), methyl orange (MO), and crystal violet (CV) dyes in both single and binary dye mixtures.A key aspect of this research is that it employs a simple and green approach to functionalize bioactive hybrid adsorbents with a high degree of efficiency. It is used to remove cationic and anionic pollutants from acidic, neutral, and alkaline solutions.

## 2. Methods

### 2.1. Materials

Raw mango seeds used in this study were generously provided by a local mango juice company. Commercial ZnO and three dyes used in this study (Figure 1)—methylene blue (MB), a cationic dye (C1_6_H_18_N_3_SCl and molar mass of 319.85 gmol^−1^), crystal violet base (C_25_H_30_ClN_3_ and molar mass of 408 g/mol) and methyl orange (C1_4_H_14_N_3_NaO_3_S and molar mass of 327.3 g/mol)—were purchased from Oxford Laboratory Reagent Company. To control the pH value, sodium hydroxide (NaOH) and nitric acid (HNO_3_) were purchased from LOBA Chemie Company, Maharashtra, India. Ethyl alcohol used for extraction was purchased from International Company, Gujarat, India. Copper(II)nitratetrihydrate (Cu(NO_3_)_2_·3H_2_O and molar mass 241.6 g/mol) was obtained from LOBA Chemie Company, India. All chemicals were used without further purification. All solutions were prepared with distilled water.

### 2.2. Preparation of Mango Seed Kernel Extract (MS-Ext)

Mango seeds (MS) were collected and washed with distilled water to remove the dust. Then, dried MS was ground and sieved by a 0.125 mm sieve. The extraction was obtained by soaking 50 g of MS powder in a 250 mL round-bottom flask with a 100 mL aqueous solution of 1:1 ethanol: water and shacked at 150 RPM for 1 h. The extract was filtrated with Whatman filter paper no. 1 and kept at room temperature before use.

### 2.3. Surface Functionalization of ZnO by the Mango Seed Extract (MS-Ext)

The surface of pristine ZnO (p-ZnO) was modified with bioactive phenolic compounds (BPCs) obtained from MS-Ext to achieve three modified samples as ZnO/MS-Ext, ZnO/MS-Ext@500 °C, and Cu-doped ZnO/MS-Ext. Three grams of ZnO powder were dispersed in 60 mL of ethanol and stirred for 15 min. 5 mL of MS-Ext was then added slowly and stirred overnight at 200 rpm. The samples were then dried overnight at 110 °C and donated to ZnO/MS-Ext. Some samples of ZnO/MS-Ext were heated at 500 °C. This sample was considered as a control to indicate the effect of the presence of phenolic and polyphenols compounds from MS-Ext on the surface of ZnO and donated ZnO/MS-Ext@500 °C.

### 2.4. Surface Functionalization of Cu Doped ZnO by MS-Ext

The synthesis and surface functionalization of Cu-doped ZnO/MS-Ext were performed directly in situ in this study. The powder of ZnO was dispersed in 60 mL of ethanol and stirred for 15 min. A specified amount of Cu^2+^ ions (in the ratio of 1:0.001) was then added dropwise and the mixture was stirred for another 15 min. Similarly, 5 mL of MS-Ext was slowly added and stirred overnight at 200 rpm. Cu-doped ZnO/MS-Ext samples were treated in the same manner as ZnO/MS-Ext samples above. ZnO surfaces containing Cu^+2^ ions may form complexes with more gallic and caffeic acid, enhancing surface negativity charge, which may enhance cationic dye adsorption rates.

### 2.5. Characterization

Surface-functionalized and commercial ZnO samples were characterized using different techniques, as follows.

#### 2.5.1. Adsorbent Surface Charge and Thermogravimetric Analysis (TGA)

Surface charge on p-ZnO, ZnO/MS-Ext, ZnO/MS-Ext@500 °C, and Cu-doped ZnO/MS-Ext was carried out by determining zeta potential using aLitesizer 500, BM10 (Anton Paar, Graz, Austria) equipped with a 40 mW 658 nm laser. All samples were measured in Omega Cuvette cells for a maximum of 300 runs at 200 volts.

TGA analysis was performed on a Discovery SDT 650 simultaneous DSC-TGA/DTA, New Castle, USA to determine the weight of the organic shell of BPCs. Typical samples weighed 4 mg and were heated in a platinum pan. Samples were heated in 20% O_2_ at a rate of 10 °C/min.

#### 2.5.2. FTIR for Detection of Functional Groups in ZnO Samples Modified by BPCs

FTIR spectra of p-ZnOand Cu-dopedZnO/MS-Ext was conducted in the 500–4000 cm^−1^ region using a NicoletiS50 spectrometer, Themo Fisher Scientific, Carlsbad, CA, USA. This system is equipped with a deuterated triglycine sulfate (DTGS) detector (Themo Fisher Scientific, Carlsbad, CA, USA) with potassium bromide windows, with a resolution of 4 cm^−1^ (45 scans). The FT-IR spectra were processed by Prestige software (IR solution, version 1.50). A cylindrical holder was used to press sample powders (200 mg, 1 wt.% in KBr) until a disk (13 mm diameter, 2 mm thickness) was obtained.

#### 2.5.3. Morphology and Elemental Analysis

A scanning electron microscope (SEM) model Quanta 250 FEG (Field Emission Gun) with an EDX unit (energy-dispersive X-ray) at an accelerating voltage of 30 kV was used to observe the morphology and the elemental analysis of the samples. Furthermore, a transmission electron microscope (TEM) model JEM-2100 Plus, was used to conduct characterizations of ZnO, ZnO/MS-Ext, andCu-dopedZnO/MS-Ext. The TEM samples were prepared by spraying an aqueous solution (0.005 weight percentage solids) onto a carbon-coated copper grid and allowing it to air-dry at room temperature.

#### 2.5.4. Adsorption and Selectivity Experiments

Effects of pristine and surfacefunctionalization adsorbents (p-ZnO, ZnO/MS-Ext, ZnO/MS-Ext@500 °C, and Cu-doped ZnO/MS-Ext) on enhancing the adsorption and the selectivity of dye molecules (with a specific charge or functional groups) wereexamined by performing selective dye adsorption experiments at a neutral pH and 25 °C. Stock solutions of different dye mixtures, namely, MB/MO (dark green), CV/MO (dark purple), and MB/CV (dark blue) were prepared by combining equal concentrations of individual dye solutions, specifically MB (blue), with MO (orange), or CV (violet), respectively. In a typical selective dye adsorption experiment, 0.005 g of adsorbent and 5 mL of dye mixture (1 g/L) were mixed in a 10 mL test tube and stirred at 200 rpm for 1.0 h. Then, the mixture was centrifuged at 3800 rpm for 5 min to separate the adsorbent from the dye solution. To study the effect of pH, experiments in a similar manner were performed at pH 4, 7, and 9. An aqueous solution of 0.1 M HCl and/or 0.1 M NaOH was used to control the solution pH using a Jenway benchtop pH meter (Jenway, Hong Kong, China). Different initial dye concentrations (10, 20, 30, 40 and 50 ppm) were performed to investigate the effect of initial dye concentration.

### 2.6. Kinetics of the Adsorption

An experiment wasconducted to determine kinetic information by adding a known amount of adsorbent to a liquid solution of a single dye and a mixed dye. The samples were collected at intervals and centrifuged for five minutes at 3800 rpm. The concentration of dye in each solution as well as the mixture solution was determined by a Jenway UV-visible spectrophotometer using a calibration curve. The absorption values were then used to calculate the quantity of the adsorbate onto the surface of adsorption at any time (*q_t_*) and at equilibrium (*q_e_*) (Equations (1) and (2)).
(1)$$q_{t}=\frac{\left(C_{0}-C_{t}\right)\times V}{m}$$
(2)$$q_{e}=\frac{\left(C_{0}-C_{e}\right)\times V}{m}$$

Perecentage removal of the pollutants was calculated according to Equation (3) [33]:(3)$$\%\;Removal=\frac{C_{0}-C_{t}}{C_{0}}\times 100$$
where *C*_0_ and *C_m_* are the concentrations of individual dye present in the dye mixture before and after adsorption, respectively, *m* (mg) is the weight of the adsorbent and *V* (L) is the volume of the solution. Then, the experimental kinetic data of $q_{t}$ and t for the selected binary system of MB/CV adsorbed onto Cu-doped ZnO/Ext were modeled according to nonlinear and linear models of pseudo-first-order, pseudo-second-order, and Elovich kinetic equations, as represented in Appendix A.

### 2.7. Adsorption Isotherm

A series of experiments wasconducted on the binary system of MB/CV at constant temperature and pH to obtain its remaining equilibrium concentration in the solution (*C_e_*) and the quantity adsorbed onto Cu-doped ZnO/Ext (*q_e_*). The information concerning the adsorption isotherm by the initial concentration of the MB/CV dye mixture was varied from 10 to 50 ppm while using a constant amount of adsorbent as 1 g/L. The obtained experimental data were modeled with Langmuir and Freundlich nonlinear forms as Equations (4) and (5), respectively [34].
(4)$$q_{e}=q_{m}\;K_{L}\;\left(\frac{C_{e}}{1+K_{L}C_{e}}\right)$$
(5)$$q_{e}=K_{F}C_{e}^{1/n}$$ Where $C_{e}$ is the [MB] and [CV] at equilibrium (mg/L), $q_{e}$ (mg/g) is the corresponding quantity adsorbed onto Cu-doped ZnO/Ext, $q_{m}$(mg/g) is the maximum adsorption capacity,$\;K_{L}$ (L/mg) is related to the energy of adsorption, $K_{F}$ (L/mg) is also related to the adsorption capacity, and n is the adsorption intensity.

## 3. Results and Discussion

### 3.1. Surface Functionalization of ZnO with MS Extract

Surface functionalization of ZnO was performed by bioactive phenolic compounds (BPCs) extracted from mango seed kernel powder (MS-Ext). The ratio of 1:1 ethanol: water was selected as a significant ratio for the solid–liquid phase extraction of BPCs from MS-Ext [35,36]. The C=O and OH groups in BPCs are used to functionalize commercial ZnO to produce different adsorbents, including ZnO/MS-Ext, ZnO/MS-Ext@500 °C, and Cu-doped ZnO-Cu/MS-Ext. The functionalized adsorbents were characterized and confirmed using various analytical techniques.

### 3.2. Characterization of Functionalized Adsorbents

#### 3.2.1. FTIR Spectra for Analyzing the Functional Groups

Figure 2a shows the FTIR spectra of p-ZnO and the surface-functionalized adsorbent of Cu-doped ZnO/MS-Ext. FTIR spectra of p-ZnO clearly displays a broad band at 3431 cm^−1^, and 1620 cm^−1^, which can be attributed to a hydroxyl group (OH) stretching and bending vibrations, respectively [37]. The bands at 1414 ${\mathrm{cm}}^{-1}$, 1034 ${\mathrm{cm}}^{-1}$, and 499 ${\mathrm{cm}}^{-1}$ were attributed to the–C–O and–C–O–C (produced during synthesis) stretching modes, as well as the Zn–O vibrations of ZnO, respectively [37]. There are also several significant peaks observed between 1300 and 1000 ${\mathrm{cm}}^{-1}$ that can be attributed to stretching vibrations in the C–O bond and bending vibrations in the O–H bond of gallic acid. On the other hand, the FTIR spectra of the functionalized adsorbent Cu-doped ZnO/MS-Ext showed a significant shift in carbonyl stretching [38,39] frequencies from 1620 to1633 ${\mathrm{cm}}^{-1}$. A significant decrease in the bands between 1300 and 1000 ${\mathrm{cm}}^{-1}$ can also be observed in Cu-doped ZnO/MS-Ext FTIR spectrum. Moreover, the increase in the broad band at 3431 ${\mathrm{cm}}^{-1}$ indicates more OH and C=O groups on the surface of Cu-doped ZnO/MS-Ext. It likely results from the interaction between phenolic compounds and ${\mathrm{Zn}}^{2+}$ ions (present on the surface of ZnO) and ${\mathrm{Cu}}^{2+}$ ions doped on the ZnO lattice of p-ZnO. According to the results, adding MS-Ext to p-ZnO powder increased the presence of OH and C=O groups on the surface. This led to the formation of a negatively charged surface that enhanced the selective adsorption of cationic organic dyes on surface-functionalized Cu-doped ZnO/MS-Ext.

#### 3.2.2. Adsorbent Surface Charge

To investigate the effect of surface functionalization using MS-Ext, the zeta potential was used to measure the surface charges of p-ZnO, ZnO/MS-Ext, ZnO/MS-Ext@500 °C, and Cu-doped ZnO/MS-Ext. The zeta potential is a physical characteristic of particles in suspension, representing electrostatic repulsion or attraction between modified adsorbents and MB, CV, and MO dyes. The electrostatic attraction may play a role in the adsorption process of the studied dyes and modified ZnO adsorbents based on the pH of the aqueous solution and the functionality of the sorbent, as will be discussed later. Figure 2b showed that p-ZnO has a positive surface charge of 3.5 mV. Upon the presence of BPCs such as gallic and caffeic acids, a negative surface charge was produced onto the surface of ZnO. The modified Cu-doped ZnO/MS-Ext exhibited the highest zeta potential of −28 mV, while the ZnO/MS-Ext exhibited a value of −13 mV. The negative surface potential of the control sample (ZnO/MS-Ext@500 °C) was reduced to −9 mV after heating to 500 °C. Clearly, ${\mathrm{Cu}}^{2+}$ ions in the modified Cu-doped ZnO/MS-Ext interacted with more phenolic compounds, resulting in an increase in surface negative charge, as indicated by the high zeta potential value. In addition, a previous study found that the dielectric constant of ZnO was decreased by doped ${\mathrm{Cu}}^{2+}$ ions in the crystal [40]. Due to this decrease in dielectric constant and the presence of negative surface charges, Cu-doped ZnO/MS-Ext can increase the adsorption of cationic dyes such as MB and CV due to a decrease in repulsion.

#### 3.2.3. Morphology and Elemental Analysis

The SEM image of pure ZnO (p-ZnO) before contacting with the solution of MS-Ext (Figure 3a) demonstrates the spherical shape and highly porous structure of ZnO (as indicated by blue arrow). Table 1 presents the results of EDX analysis of p-ZnO (Figure 3b) that confirms the presence of elemental zinc (Zn) and oxygen (O) signals only with 81.13% (mass%) zinc (Zn) and 18.87% oxygen (O) atoms. In the SEM image (Figure 3c), the functionalization of the ZnO surface is evident by the agglomeration and the dense shape of ZnO due to the presence of bioactive phenolic compounds (BPCs) in the interspace of ZnO as indicated by the green arrow. The results of the EDX analysis (Figure 3d) further confirmed the presence of 12.70% carbon (C) atoms that became apparent after BPC functionalization. Furthermore, the SEM image (Figure 3e) clearly illustrates the presence of Cu ions in the interspace of ZnO (as indicated by the red arrow). EDX analysis (Figure 3f) provided direct evidence for the presence of 7.22% carbon (C) from BPC extract and 1.23% ions doped into ZnO/MS-Ext particles as shown in Table 1. It is possible that BPCs and Cu^2+^ ions played a significant role in the enhanced adsorption of organic dyes in aqueous solutions. Furthermore, TEM was used to further observe the morphology of the modified adsorbents. As shown in Figure 4a–c, TEM images depict morphological properties in terms of the sizes and shapes of p-ZnO, ZnO/MS-Ext, and Cu-doped ZnO/MS-Ext. P-ZnO particles tend to form subtle clusters of different shapes, including tetrahedral, rod-shaped, cube-shaped, and spherical particles. The particle sizes ranged between 111 and 1430 nm, with an average size of 441 nm. As a result of the modification of p-ZnO with BPC, the average particle size of ZnO/MS-Ext was significantly reduced by 58% to 184 nm compared to 441 nm for p-ZnO. A further significant reduction in particle size was achieved by the presence of Cu^2+^ ions (as indicated by red arrow) in Cu-doped ZnO/MS-Ext (Figure 4e), from 441 nm to 237 nm. Although both adsorbents were shaped like cubes and rods, the small particles of Cu-doped ZnO/MS-Ext (see Figure 5) had a flower-like outline. The SEM and TEM images, the size distribution, FTIR, and zeta potential of ZnO/Ext and Cu-doped ZnO/MS-Ext werecompared with those of ZnO. The results demonstrated that p-ZnO was successfully functionalized with MS-Ext to form modified adsorbents with negatively charged surfaces, which enhanced organic dye adsorption and selectivity in single and binary systems.

### 3.3. The Role of MS-Ext in Enhancing Adsorption

During the functionalization process, MS-Ext was added to the powder of p-ZnO, resulting in a brownish dispersion indicating the presence of various interactions with BPCs. As reported previously [35], MS-Ext contains a high concentration of gallic acid (GA) and caffeic acid (CA), as shown in Figure 5. Thus, it is possible to propose surfacefunctionalization of ZnO with polar BPCs such as gallic acid with four protons (H_4_A) and caffeic acid with three protons (H_3_A). There are different functional groups present in gallic acid and caffeic acid, including hydroxyl (OH), carbonyl (C=O), phenolic, benzene ring and carboxylic (COOH) groups.

The C=O in gallic acid and caffeic acid possesses a nucleophilic property and can form complexes with transition metals such as ${\mathrm{Zn}}^{2+}$(on the surface of ZnO) and ${\mathrm{Cu}}^{2+}$ ions in Cu-doped ZnO/MS-Ext.Then surface-functionalized gallic acid and caffeic acid (Figure 5) may undergo further protonation/deprotonation of OH and COOH groups to produce negatively charged phenolate ions [41], as shown in Equation (6). It was reported that the degree of deprotonation (DoD), or the ratio of gallic acid to gallate anions, is affected by the pH of the solution [41]. Therefore, gallic acid possesses four pKa values (Equation (6)).
(6)$$H_{4}A+\begin{array}{c} H_{2}O\\ ↔\\ pK_{a_{1}=4.39} \end{array}\;H_{3}A^{-}\begin{array}{c} H_{2}O\\ ↔\\ pK_{a_{2}=8.5} \end{array}H_{2}A^{2-}\begin{array}{c} H_{2}O\\ ↔\\ pK_{a_{3}=10.38} \end{array}\;\;\;\;\;HA^{3-}\begin{array}{c} H_{2}O\\ ↔\\ pK_{a_{4}=13} \end{array}A^{4-}$$

At low pH of aqueous solution, gallic acid with four protons (*H_4_A*) is the dominant species. As the pH increases, the relative deprotonation of $H_{4}A$ increases until almost all protons are donated, where the predominant species in the solution are $HA^{3-}\;\mathrm{and}\;H_{2}A^{2-}$ at pH of 10–11 (Equation (6)). These species can participate in enhancing both the selectivity and adsorption process. According to previous research, gallic acid interactions enhance the surface reactivity of clay particles coated with iron oxides [42].

This suggests that functionalized adsorbents (ZnO/MS-Ext and Cu-doped ZnO/MS-Ext ZnO) with H_4_A and H_3_A would exhibit an increase in surface negative charge and acid character with an increase in solution pH [43], which would increase the affinity for MB and CV dyes. We conducted several adsorption experiments to evaluate ZnO/MS-Ext and Cu-doped ZnO/MS-Ext for enhancing the removal of MB, CV, and MO dyes in single and binary dye mixtures.

### 3.4. Adsorption of Single and Binary System onto p-ZnO

The efficiency of pristine ZnO (p-ZnO) for adsorption of MB was studied. Screening adsorption experiments of a single system of aqueous solutions of 20 ppm MO, MB, and CV and binary systems of MB/CV and MB/MO were investigated at natural pH (pH 7.5).

The results in Figure 6a show that p-ZnO was not efficient in the single system, removing only 7.1%, 7.27% and 16.43% of MB, CV and MO, respectively. As indicated in the inset in Figure 6a, both the white color of the p-ZnO powder and the original color of the dye solutions changed slightly in 60 min at pH 7.5.

#### The Influence of Competitive Adsorption on Selectivity onto p-ZnO

An evaluation of the effects of different interactions and competing adsorptions of MB/MO (anionic/cationic system) and MB/CV (cationic/cationic system) onto the surface of p-ZnO was conducted at different pH values (pH 4.4, pH 7.5, and pH 10.7) and the results are shown in Figure 6. Experimental performed with a concentration ratio of 1:1 (20 ppm for each dye). Hence pure ZnO possesses different surface charges at different pH levels. The comparative adsorption in both systems was vary by combining both dyes in the solution at different pH. At pH 7.5, it was found that the addition of the anionic dye MO to MB solution led to a slight increase in the percentage removal of MB from 7.1% in the single system to 28% in the binary system of MB/MO as shown in Figure 6b. Meanwhile, the presence of MB resulted also in a slight increase in the percentage of MO removed from 16.4% to 27.5%, whereas the presence of MO dye in the mixture of MB/MO in acidic and alkaline media led to a reduction in the competitive adsorption of both MB and MO on the surface of p-ZnO. Figure 6b shows that the percentage removal of MB decreased to 27.1% and 21.27% at pH 4.4 and pH 10.7, respectively. In addition, the removal of MO decreased to 6.11% and 6.71% at pH 4.4 and pH 10.7, respectively. In the binary system of MB/CV, mixing cationic dye CV with cationic dye MB resulted in a dramatic decrease in the percentage removal of both MB and CV in acidic medium and a slight increase in alkaline medium, as shown in Figure 6b. At pH 4.4, the MB removal decreased from 2.36% to 0.32% and the CV from 2.88% to 1.16%, while at pH 10, the MB removal increased slightly to 5.23% and the CV increased to 9.35%. These results can be attributed to the surface charge of ZnO and the pK_a_ of each dye. The zeta potential was determined to examine the surface of p-ZnO (Figure 2b) and showed that the surface of p-ZnO is positive at natural pH (pH 7.2) with a value of 3.5 mV. Furthermore, our previous investigations [34] among other reports [11,44] have shown that p-ZnO possesses a zero point of charge (pH_pzc_) of 7.3 to 8.1. At pH 7.2< pH_pzc_, the surface of p-ZnO is positively charged and leads to electrostatic repulsion and electrostatic attraction due to the nature of the binary system MB/MO. The pK_a_ values were also given as 3.46, 3.14 and 0.8 for MO, MB and CV, respectively [45,46]. The MO dye is primarily present in the azo structure as an anionic form at pH 7.2 (pH > pk_a_), as shown in Figure 1. This resulted in enhanced adsorption on the positively charged p-ZnO. Accordingly, the presence of MO on the surface increases the negative charge and speeds up the competitive adsorption of the cationic MB dye to 28.3%. On the other hand, in the MB/CV mixture, both MB and CV are ionized at pH 7.2 and pH 4.4 (pH > pK_a_) and exist as cationic species. Therefore, the adsorption rate of both MB and CV is dramatically reduced due to electrostatic repulsion with the positively charged p-ZnO surface in addition to competition with H^+^ ions at pH 4.4. As pH increased in an alkaline medium at pH 10.7, the electrostatic attraction was enhanced and led to an increase in the adsorption of positively charged MB dye [47]. In contrast, the presence of cationic MB dye on the surface of ZnO enhances the adsorption of anionic MO dye by 9.36%. Based on the above results, p-ZnO did not provide effective adsorption for the three dyes MB, MO, and CV in either single or binary systems. Consequently, it is challenging to modify the surface of p-ZnO to increase the rate of adsorption.

### 3.5. Effect of SurfaceFunctionalization of p-ZnO on Enhancing Adsorption Process

As discussed previously, the negatively charged surface of the functionalized adsorbents of ZnO/MS-Ext, and Cu-doped ZnO/MS-Ext may enhance the adsorption of MB, MO, and CV dyes in single and binary systems through hydrogen bonds, electrostatic interactions, π–π interactions (with gallate rings) and coordination bonds with ${\mathrm{Cu}}^{2+}$ ions.

#### 3.5.1. Adsorption of Single and Binary Systems onto ZnO/MS-Ext and ZnO/MS-Ext@500 °C

As shown in Figure 7a, in the single system of MB, CV and MO dyes at pH 7.5, ZnO/MS-Ext significantly enhanced the selectivity (removal) of cationic dyes MB and CV from 7.1% and 7.3% by p-ZnO to 75.45% and 67.2%, respectively. In contrast, the selectivity of anionic dye MO decreased from 16.4% to 6.6%. The results can be attributed to the formation of negatively charged gallate ions via the functionalization of p-ZnO with gallic acid and caffeic acid from MS-Ext, as discussed previously. Further evidence of the role of the negatively charged gallate ions can be supported by the weight loss, as shown in the TGA results (Figure 7b). The melting points of gallic acid and caffeic acid are 213 °C and 260 °C, respectively. At 500 °C, the two acids melt and are removed from the surface, resulting in a decrease in the negative charge of gallate ion on the surface of ZnO/Ext. The results indicated that functionalized ZnO/MS-Ext adsorbent exhibited small losses (0.34% to 1.86%) at temperatures between 100 °C and 200 °C, which may be attributed to the loss of adsorbed water and small organic molecules formed from the MS extract. It is thought that the large mass loss of 9.7% (about 0.71 mg) is a result of the decomposition of gallic acid and caffeic acid, which are formed during the functionalization process of ZnO/MS-Ext. It was noted that mass loss ends at temperatures above 500 °C. Due to the decomposition of both gallic acid and caffeic acid during the calcination process, there was no significant weight loss in the calcined ZnO/MS-Ext @500 °C when compared to the uncalcined ZnO/MS-Ext. Accordingly, the selectivity and thus the removal of three dyes by ZnO/MS-Ext @500 °C was significantly decreased to 1.73%, 24.57% and 23.06% for MO, MB, and CV, respectively, at pH 7.5, due to the decomposition of gallate ions, as shown in Figure 7a.The formation of the negative charges by gallate ions on the surface of ZnO/MS-Ext is thought to play a key role in the enhancement of the adsorption process of cationic dyes through electrostatic interactions. It was noted that the adsorption of MO dye at pH 7.5 onto the surface of ZnO/MS-Ext at 500 °C was dramatically decreased to 1.7. At pH 7.5, which is in the range of pH_pzc_, the surface of ZnO/MS-Ext @500 °C exhibits no charge and thus has less affinity to adsorb anionic MO dyes.

Figure 8a–c shows the removal of the binary systems of MB/CV, MB/MO and CV/MO onto the surface of ZnO/MS-Ext at different pH values (4.4, 7.5, and 10.7). The results indicated that the competitive adsorption significantly influenced the selectivity of the three binary systems. In the binary of the cationic dyes MB/CV (Figure 8a), the removal of MB was 33%, 23% and 52%, respectively, and removal of CV was 48%, 34%, and 57% at pH 4.4, 7.5 and 10.7, respectively. Due to the competition of the two dyes for the same sites, the removal of MB and CV in MB/CV was diminished when compared to the single system. However, the decrease in the removal of the two dyes was affected by the solution pH. This may attribute to the pK_a_ values of gallic acid (4.4, 8.5 and 10.4), which may vary the presence of neutral and ionic species of gallate ions on the surface of the adsorbent. At pH 4.4, there is less negative charge on the surface in addition to the competition of H^+^ ions with cationic dyes MB and CV, resultingin decreased removal [48,49]. At pH 7.5 (pH_pzc_), the surface exhibited neutral charge, resultingin low affinity to cationic dyes. In addition, increasing the pH to pH 10.7 increased the surface negative charge and enhanced theremoval to 52% and 57% for MB and CV, respectively. On the other hand, the presence of the anionic dye MO in the binary system of MB/MO led to enhancing the adsorption of MB dye at pH 4.1 and pH 7.5 compared to the cationic binary MB/CV. At pH 4.4, the negative charge produced from the ionization of OH of the COOH of gallic acid enhanced the adsorption of MB due to the absence of the competition from the anionic MO dye, although at pH 7.5, the surface of ZnO/Ext is relatively neutral (at pH_pzc_), theremoval increased to 68% and 58% for MB and MO, respectively. This result can be explained by the interaction with the surface of ZnO/Ext through π–π interactions [50,51]. In contrast, at pH 10.7, theremoval decreased from 68% to 51% for MB dye and dramatically reduced from 58% to 9% for MO dye. The electrostatic repulsion of the anionic MO dye along with the competition with OH^−^ ions in the alkaline solution can play a key role indecreasing the adsorption of MO. The competition with OH^−^ for the active sites may also decrease the adsorption of MB dye. The results were further proved by the FTIR spectra of the three binary systems after adsorption onto the surface of ZnO/Ext along with the spectra of pure ZnO/Ext, as shown in Figure 8d. There was a significant change observed in the bands between 1603 cm^−1^ and 1000 cm^−1^ of the stretching vibrations in the OH and C=O groups and the stretching vibrations in the C–O bond and bending vibrations in the O–H bond of gallic acid. As can be seen, the binary MB/MO showed a change in the bands between 1603 and 1000 cm^−1^ that was more significant than the binary of MB/CV and CV/MO.

#### 3.5.2. Adsorption of Single and Binary Systems onto Cu-Doped ZnO/MS-Ext

Figure 7a representsthe comparison between the adsorption of single MB, CV, and MO dyes onto the surface of p-ZnO, ZnO/MS-Ext and Cu-doped ZnO/MS-Ext. It can be seen that Cu-doped ZnO/MS-Ext exhibited the highestremoval of the three dyes at pH 7.5. The adsorbent selectively adsorbed cationic dyes MB and CV with 98.3% and 93.7% respectively. However, the anionic MO dye showed the lowest selectivity with 11.2% removal. The results can be attributed to the increase the negativity of the surface of Cu-doped ZnO/MS-Ext due to the contribution of Cu^2+^ ions in the coordination interactions with more gallic and caffeic acids. In addition to the electrostatic interactions of MB and CV dyes with the negative surface, coordination bonds and π–π interactions may contribute to the adsorption process. In contrast, the electrostatic repulsion decreases the adsorption of the anionic dye MO onto the surface of the adsorbent. On the other hand, Figure 9a–c illustrates the adsorption of the binary systems MB/CV, MB/MO, and CV/MO on Cu-doped ZnO/MS-Ext. All three binary systems displayed significant differences in selectivity due to competitive adsorption. In acidic and alkaline media, both MB and CV were significantly removed. It was found that CV dye was removed more effectively (with 92% and 87%) than MB dye (with 72% and 87%), in acidic and alkaline media, respectively. The presence of anionic MO dye in the binary MB/MO resulted in a significant increase in the selectivity of MB dye, with 84% and 92% removal in acidic and neutral media, respectively. Conversely, MO dye decreased theremoval of MB to 57% in an alkaline medium. However, the adsorption of the anionic MO dye onto the surface of Cu-doped ZnO/Ext was significantly inhibited by1.1%, 44%, and 16% in acidic, neutral, and alkaline media respectively. Figure 9a,c illustrate a comparison of CV adsorption in binary systems of MB/CV and CV/MO. In the binary system MB/CV (Figure 9a), Cu-doped ZnO/Ext demonstrated a high affinity towards CV dye, as shown by 92% removal for CV dye, compared to 72% removal for MB dye. The percentage removal for MB and CV decreased to 36% and 57%, respectively, in a neutral solution at pH 7.5, which is very close to the point of zero charge (PZC) of ZnO. At PZC, the adsorbent surface is neutral, which may decrease the affinity for cationic dyes. On the other hand, at pH 10.7 the surface of the adsorbent is highly negative, which enhances the adsorption of cationic dyes, as shown in Figure 9c.However, the adsorption of CV dye decreased to 81% and 59% in acidic and alkaline media, respectively, while increasing to 65% at pH 7.5.The results can be attributed to a variety of interactions such as electrostatic attraction and repulsion, coordination bonds, and π–π interactions, as discussed previously.

### 3.6. Adsorption Kinetics of MB/CV Binary System

In an adsorption process, adsorbates such as MB, CV and MO are transferred from the fluid phase to the surface of the functionalized ZnO. It is desirable to investigate the kinetics of adsorption to gain insight into the mechanism of adsorption, which plays an important role in the efficiency of the process, to avoid negative effects resulting from the conversion of nonlinear regression into the linear form of most kinetic models [34].

In this study, we examined the kinetics of MB/CV binary systems using both nonlinear and linear models of pseudo-first-order, pseudo-second-order and Elovich equation [52] for the concentration range from 10 to 50 ppm. The equations of both linear and nonlinear forms are given in Appendix A. The selected model was chosen based on two statistical error analysis functions as the residual sum of squared errors (RSS), and coefficient of determination ($R^{2}$) based on function builder of Origin software (OriginLab Corporation, Northampton, USA). For each dye of MB and CV in MB/CV mixture, the experimental values of qt were ploted versus time (*t*) to determine the values of $k_{1},\;q_{e}\mathrm{and}R^{2}.$ Figure 10a,b represented the plot of the nonlinear forms of PFO, PSO for MB and CV in MB/CV mixture respectively. For comparison, the plot of linear forms of PFO and PSO for MB and CV in MB/CV mixture are presented in Appendix A, respectively. Note that the calculated parameters as well as statistical error analysis functions for the nonlinear forms of PFO and PSO are presented for MB and CV in Table 2 and Table 3, respectively. While the calculated parameters for the linear forms of PFO and PSO are shown for MB and CV in Appendix A, respectively. As shown in Figure 10a and Table 2, the adsorption kinetics of the MB dye in the binary system MB/CV suggest that the nonlinear form of the PSO model has a high $R^{2}$ (0.990–0.999) and a high RSS (0.42–1.74). Furthermore, there is good agreement between the experimental and calculated values of $q_{e}$. However, when the experimental data were converted to the linear form of the PSO model, the $R^{2}$ values were higher (0.996–1.000), and the RSS values were smaller (0.003–0.403). Furthermore, the experimental and calculated values of $q_{e}$ were in good agreement. Consequently, the kinetics of the MB dye in the mixture followed the linear form of the PSO model. It appears that the conversion of experimental data to linear form results in a positive effect on the determination of the kinetic parameters.

Similarly, Figure 10b represents the plot of the nonlinear forms of PFO and PSO for CV in MB/CV mixture. The results (Table 3) showed that in comparison with PFO, the linear model for PSO had a higher $R^{2}$ (0.994–1.00) and the lowest RSS values (0.002–1.044). Also, there is good agreement between the experimental and calculated values of $q_{e}$. As indicated by the kinetic data, the chemisorption process may also be contributing to the determination of the rate step. It is thought that electrons are shared between functional groups of BPCs on the surface of adsorbents and those of MB and/or CV dyes, where valency forces are present [53].

In addition, experimental data of the adsorption kinetics of MB and CV in MB/CV mixture were treated using both linear and nonlinear Elovich models (Equations are shown in Appendix A). The plot of the nonlinear form is shown in Figure 10c,d, and the plot of the linear form is shown in Appendix A for MB and CV, respectively. Note that for the comparison, the calculated parameters for both linear and nonlinear are shown in Table 4. The results indicated that the values of β (desorption constant) for MB in MB/CV mixture obtained from the nonlinear form (0.152–0.889 mg/g) and linear form (0.117–0.585 mg/g) are both very small values and similar indicating an irreversible process. Nevertheless, the calculated error analysis indicated that the $R^{2}$ values for the nonlinear ranged from (0.976–0.999) are greater than those for the linear form (0.929–0.975). Additionally, the RSS values for the nonlinear form (0.052–5.78) are significantly smaller than those for the linear form (1.14–82.48). Thus, the nonlinear model of Elovich is suitable for modeling MB dye adsorption onto Cu-doped ZnO/Ext. On the other hand, the kinetic data for CV dye in MB/CV mixture (Table 5) showed that the linear form of Elovich was significantly more suitable due to higher $R^{2}$ values (0.950–0.993) and lower RSS values than nonlinear form. The results also indicate that the kinetic parameters of the adsorption process cannot be determined by converting the data of the nonlinear form to the linear form. The best-fitting and error analysis functions indicate that the kinetics of the adsorption of MB onto Cu-doped ZnO/Ext follow the order linear PSO > nonlinear PSO > nonlinear Elovich model > linear Elovich model. In contrast, the kinetics of the adsorption of CV dye adsorption followed the linear PSO > nonlinear PSO > linear Elovich > nonlinear Elovich model.

### 3.7. Adsorption Isotherm of MB/CV Binary System

Using adsorption isotherms allows us to gain valuable insight into the mechanism of adsorption (monolayer or multilayer) as well as the type of interaction that occurs during adsorption (physical or chemical adsorption) in addition to investigating the efficiency of a design adsorption system. To investigate the performance of Cu-doped ZnO/Ext for the removal of MB/CV mixture, the experimental data were treated with the nonlinear forms of Langmuir and Freundlich isotherm (Equations (3) and (4)). Residual sum of squared errors (RSS) and coefficient of determination ($R^{2}$) (Equations (3) and (4)) were used to select the best-fit model. Figure 11a,b shows the relation between the experimental data and the Langmuir and Freundlich models for the adsorption isotherm of MB and CV in MB/CV mixture onto the surface of Cu-doped ZnO/Ext. The calculated parameters along with the statistical error functions are shown in Table 6. A Langmuir model is shown to better describe the adsorption isotherm of both the MB and CV dyes in the mixture than a Freundlich model. For MB and CV, the values of $R{\;}^{2}$ calculated from the Langmuir isotherm (0.982 and 0.998) were higher than those obtained from the Freundlich model (0.954 and 0.987). In addition, this result was supported by the relatively low RSS values (2.14 and 22.38) compared to the high RSS values (35.88 and 4.84) for the Freundlich model. According to the Langmuir model, this result suggests that a monolayer of MB/CV mixture is chemisorbed on the surface of Cu-doped ZnO/Ext at a finite number of definite localized sites rather than occurring in multiple layers [54].

This may be explained by the ionic and coordination interactions of the MB/CV mixture and the localized adsorption active centers of the functional groups (C=O and OH) and $Cu^{2+}$ present on the functionalized Cu-doped ZnO/Ext. The maximum adsorption capacity of Cu-doped ZnO/Ext in the MB/CV binary system was found to be 46.6 mg/g for MB and 72.5 mg/g for CV. In addition, the calculated Langmuir constant KL for CV and MB was 0.836 and 0.077 L/mg, respectively. The increasing value of Kl indicates that CV has a strong affinity for the surface of Cu-doped ZnO/Ext.

### 3.8. Adsorption Mechanism

Figure 12 illustrates the main adsorption mechanism of MB, CV, and MO dyes in a single and binary system onto a Cu-doped ZnO/Ext surface. An increase in negative charge is observed on a Cu-doped ZnO surface-functionalized with MS-Ext (Figure 12a), as demonstrated by an increase in zeta potential from 3.5 to −28 eV. The H-bonding between N atoms of amino groups and N atoms of OH groups in GA and CA may allow cationic dyes like MB and CV to adsorb onto the surface of doped Cu-doped ZnO/Ext (Figure 12b). As discussed previously, π–π interactions may also contribute to adsorption. As shown in Figure 12c, electrostatic interactions between cationic MB and CV, along with π–π interactions, may enhance the adsorption of anionic MO dye in the binary system of MB/MO. Adsorption kinetics suggest that a second-order model represents the adsorption process. In addition, the Langmuir adsorption isotherm indicates that MB, CV, and MO dyes were chemisorbed on the surface of Cu-doped ZnO/MS-Ext as monolayers. It is possible that the functional groups (C=O and OH) present on the functionalized Cu-doped ZnO/Ext from MS-Ext act as active centers for dye adsorption. In conclusion, electrostatic and coordination interactions as well as π–π interactions determine the adsorption mechanisms of MB, CV, and MO dyes.

## 4. Conclusions

Using MS-Ext to functionalize commercial ZnO resulted in the development of an effective adsorbent for the removal of complex organic dye mixtures. It was confirmed by the FTIR spectrum that MSextract interacts with p-ZnO. In single and binary systems, Cu-doped ZnO/Ext can adsorb MB, CV, and MO dyes more effectively. A binary system consisting of MB/CV showed a maximum adsorption capacity (q_m_) of 72.49 and 46.61 mg/g, respectively. For the functionalized adsorbents, the zeta potential increased from 3.5 eV for p-ZnO to −28 eV for Cu-doped ZnO/MS-Ext. In this study, the adsorption process was modeled using a second-order model and Langmuir adsorption isotherm. These results indicate that MB, CV, and MO dyes are chemisorbed onto adsorbent surfaces as monolayers at localized active centers of MS-extract functional groups. Adsorption of cationic and anionic dyes onto Cu-doped ZnO/Ext surfaces is the result of electrostatic attraction/repulsion, coordination bonds, and π–π interactions. Moreover, the use of MS-Ext for the functionalization process is cost-effective and can help reduce solid waste in the environment.

In summary, a simple and eco-friendly approach is highlighted to functionalize bioactive hybrid adsorbents with high efficiency. The results of this study indicate that Cu-doped ZnO/Ext is an effective alternative adsorbent for the removal of MB, CV, and MO from aqueous solutions in a single and binary system.

## Figures and Tables

**Figure 1 ijerph-20-05750-f001:**
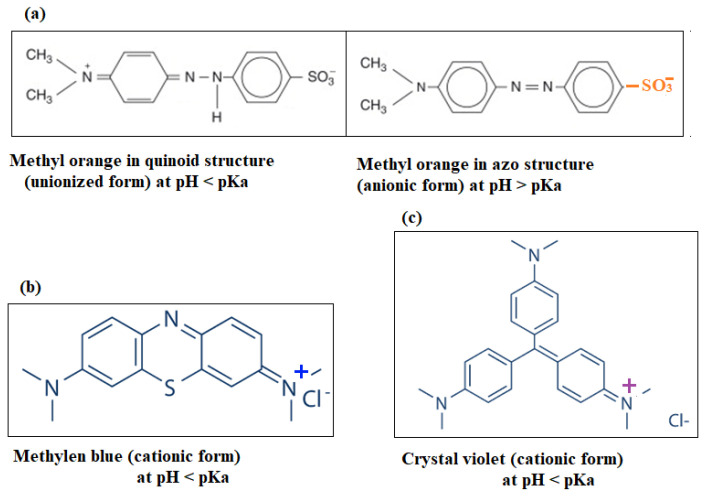
Chemical structure of studied dyes (**a**) Methyl orange in quinoid structure (unionized form) and methyl orange in azo structure (anionic form)*,* (**b**) methylene blue (cationic form), and (**c**) crystal violet (cationic form).

**Figure 2 ijerph-20-05750-f002:**
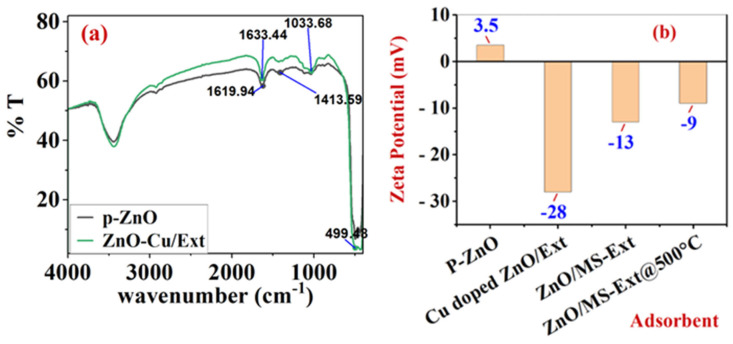
(**a**) Infrared spectra of p-ZnOand Cu-dopedZnO/MS-Extand (**b**) zeta potential values of adsorbents.

**Figure 3 ijerph-20-05750-f003:**
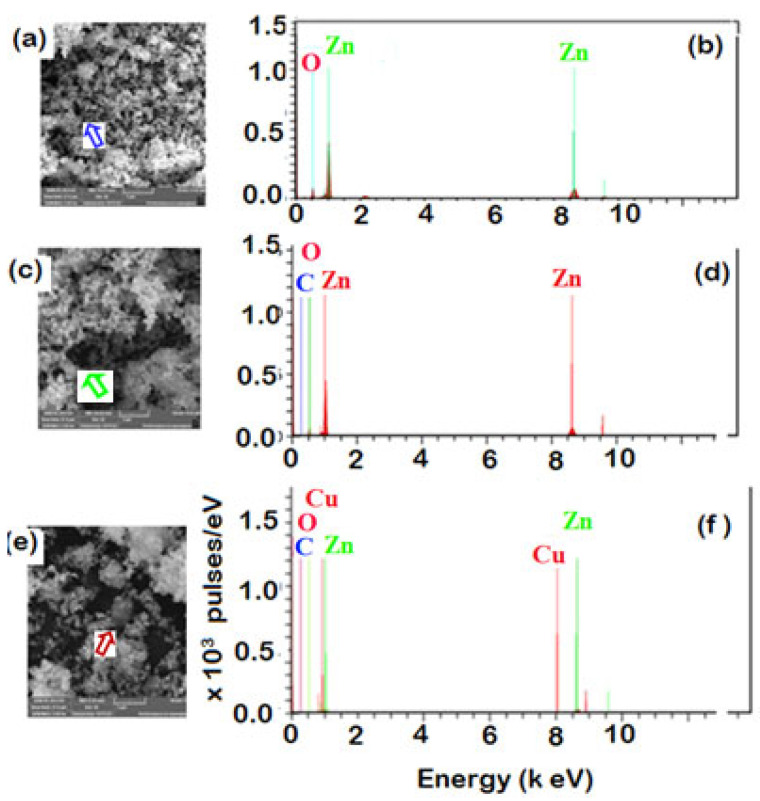
EDX and SEM images for ZnO surface: (**a**) SEM image of p-ZnO, (**b**) EDX analysis of p-ZnO, (**c**) SEM image of ZnO after surface functionalization with MS-Ext (**d**) EDX analysis of ZnO after surface functionalization with MS-Ext, (**e**) SEM image of ZnO after surface functionalization with MS-Ext and doping with Cu^2+^ and (**f**) EDX analysis after surface functionalization with MS-Ext and doping with Cu^2+^.

**Figure 4 ijerph-20-05750-f004:**
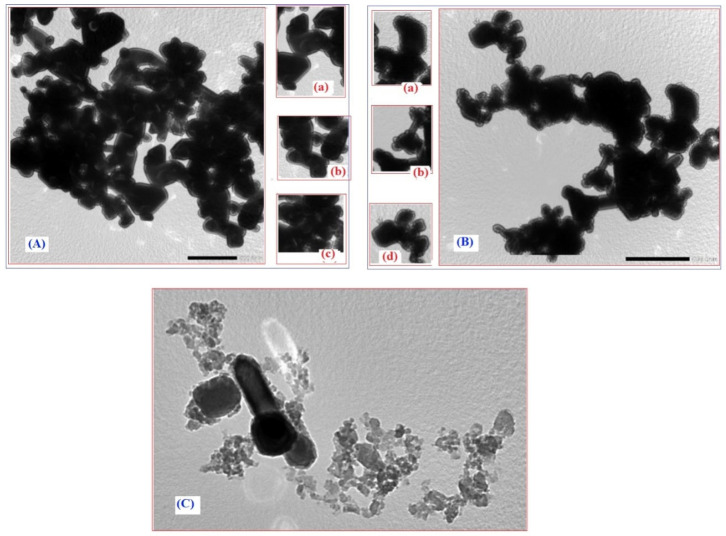
TEM images of (**A**) p-ZnO, (**B**) ZnO/MS-Ext, and (**C**) Cu-doped ZnO/MS-Ext. The enlarged views show particles of (**a**) tetrahedral shape for p-ZnO, (**b**) rod shape for ZnO/MS-Ext, (**c**) spherical shape and (**d**) cube-shapefor Cu-doped ZnO/MS-Ext.

**Figure 5 ijerph-20-05750-f005:**
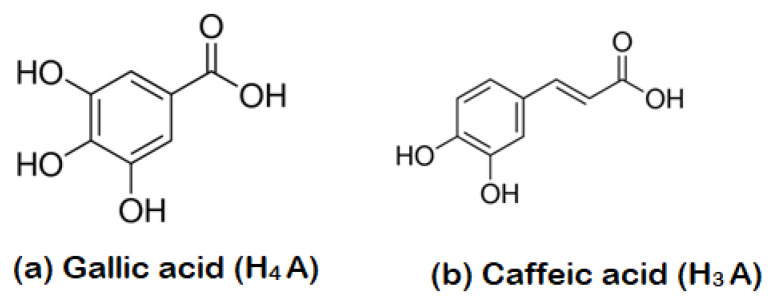
Chemical structure of (**a**) gallic acid (H_4_A) and (**b**) caffeic acid (H_3_A) in MS-Ext.

**Figure 6 ijerph-20-05750-f006:**
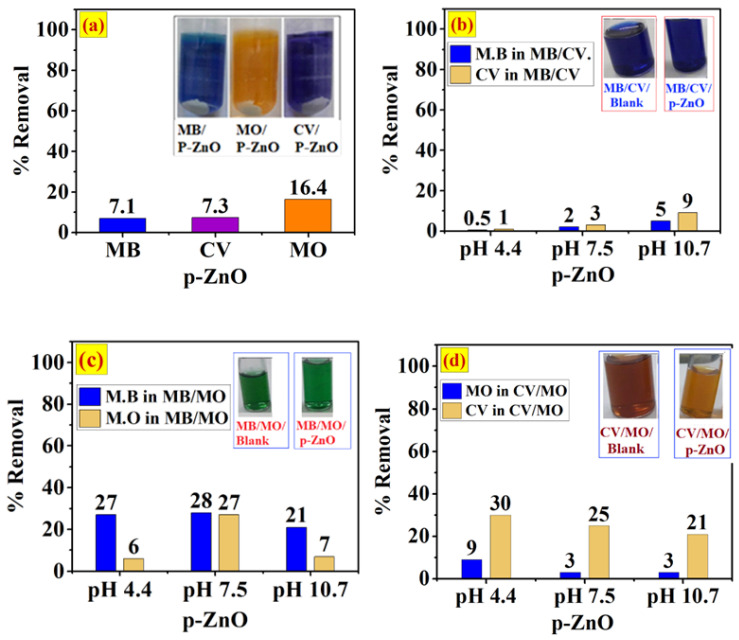
Adsorption of single MB, CV, and MO dyes at pH 7 (**a**), binary MB/CV (**b**), binary MB/MO (**c**), and binary CV/MO (**d**) onto the surface of p-ZnO at pH 4.4, 7.3, and 10.7. The inset shows a photograph of the single and binary dye solution compared to the blank.

**Figure 7 ijerph-20-05750-f007:**
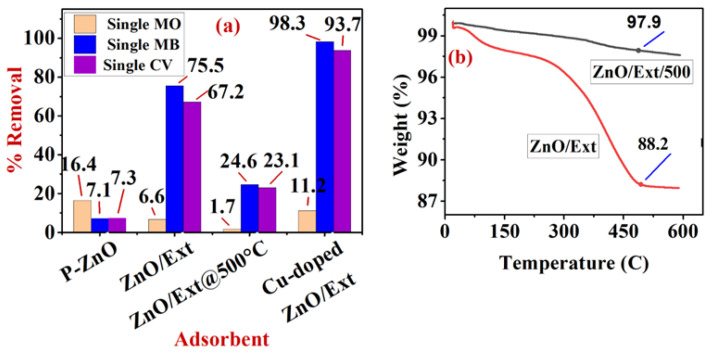
Adsorption of single MO, MB, and CV dyes onto the surface of functionalized adsorbents p-ZnO, ZnO/MS-Ext, ZnO/MS-Ex@500 °C and Cu-doped ZnO/Ext at pH 7.5 (**a**) and TGA thermogram curve for ZnO/MS-Ext, and (**b**) ZnO/MS-Ex@500 °C.

**Figure 8 ijerph-20-05750-f008:**
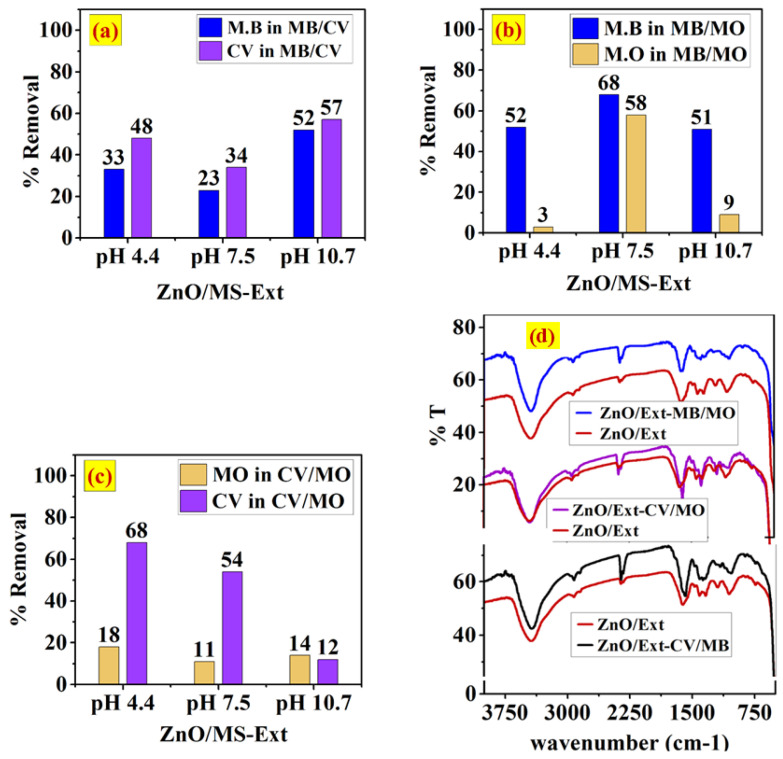
Adsorption of the binary system of (**a**) MB/CV, (**b**) MB/MO and (**c**) CV/MO dyes onto the surface of ZnO/Ext at pH 4.4, 7.3, and 10.7, and (**d**) FTIR spectra of the binary systems after adsorption compared to the pure ZnO/Ext.

**Figure 9 ijerph-20-05750-f009:**
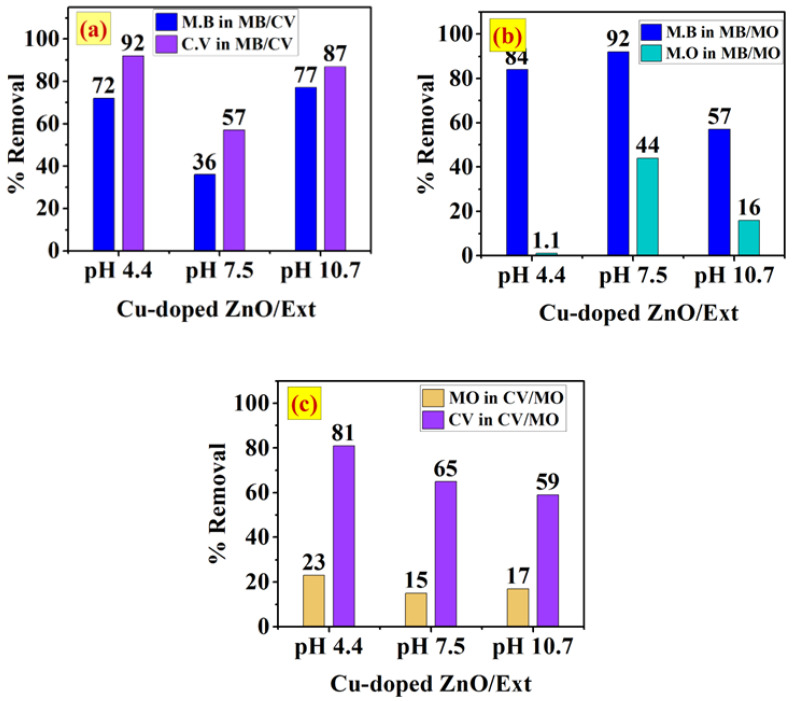
Adsorption of the binary system of (**a**) MB/CV, (**b**) MB/MO and (**c**) CV/MO dyes onto the surface of Cu-doped ZnO/Ext at pH 4.4, 7.3, and 10.7.

**Figure 10 ijerph-20-05750-f010:**
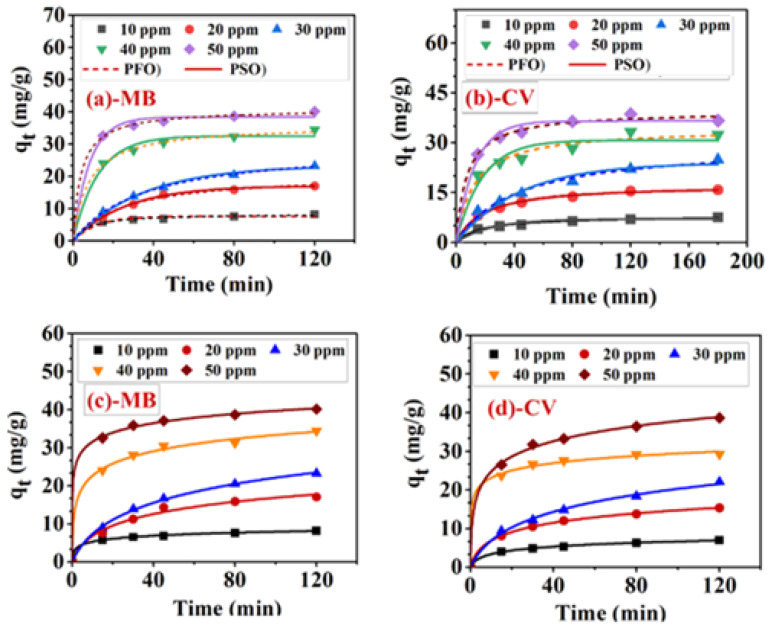
The nonlinear curves of PFO and PSO kinetics of adsorption of (**a**) MB in MB/CV, (**b**) CV in MB/CV, Elovich model for (**c**) MB in MB/CV and (**d**) CV in MB/CV binary system using Cu-doped ZnO/Ext.

**Figure 11 ijerph-20-05750-f011:**
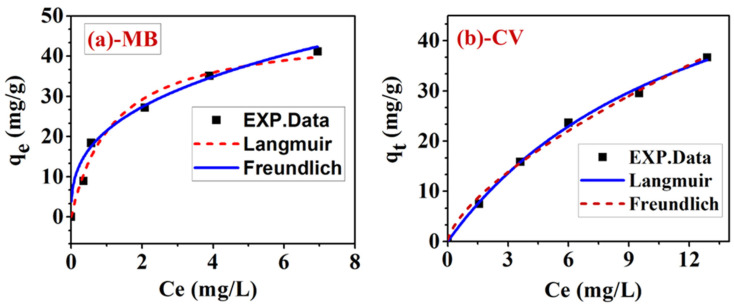
Langmuir and Freundlich nonlinear forms of adsorption isotherm for the adsorption of (**a**) MB dye and (**b**) CV dye in MB/CV binary system onto Cu-doped ZnO/Ext at pH 7.5 and at room temperature.

**Figure 12 ijerph-20-05750-f012:**
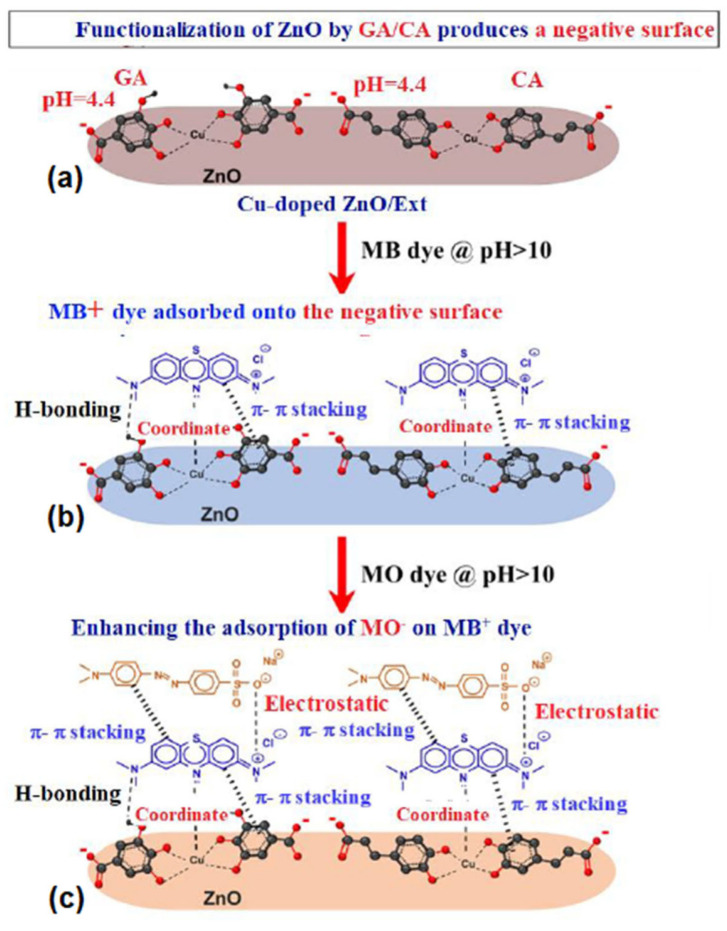
The adsorption mechanism of MB, CV and MO dyes onto Cu-doped ZnO/MS-Ext. (**a**) Functionalization of ZnO by GA/CA produced a negatively surface charged ZnO, (**b**) Adsorption of cationic ${\mathrm{MB}}^{+}$ dye onto negative surface charged ZnO, and (**c**) Cationic ${\mathrm{MB}}^{+}$ enhanced adsorption of anionic ${\mathrm{MO}}^{-}$ dye.

**Table 1 ijerph-20-05750-t001:** EDX weight ratio of electrospun for p-ZnO, ZnO/MS-Ext and Cu-doped ZnO/MS-Ext.

	p-ZnO		ZnO/MS-Ext		Cu-doped ZnO-MS-Ext	
Element	Mass [%]	Atomic [%]	Mass [%]	Atomic [%]	Mass [%]	Atomic [%]
Zinc	81.13	51.27	66.91	30.50	85.53	56.76
Oxygen	18.87	48.73	20.40	38.00	6.02	16.32
Carbon	0	0	12.70	31.50	7.22	26.08
Copper	0		0	0	1.23	0.84
Sum	100		100	100	100	100

**Table 2 ijerph-20-05750-t002:** Nonlinear parameters of pseudo-first-order and pseudo-second-order for the adsorption of MB dye in MB/CV binary system onto Cu-doped ZnO/Ext at pH 7.5 and at room temperature.

Nonlinear Pseudo-First-Order						Nonlinear Pseudo-Second-Order			
[MB] ppm	$q_{e,exp\;}$ (mg/g)	$q_{e,cal\;}$ (mg/g)	$k_{1\;}$ (min^−1^)	$R^{2}$	RSS	*q_e. (cal)_* (mg/g)	$k_{2}$ (min^−1^)	$R^{2}$	RSS
10	8.55	7.88	0.074	0.965	0.492	8.64	0.0135	0.990	0.492
20	17.2	17.10	0.038	0.998	1.74	19.94	0.0023	0.993	1.738
30	23.33	23.41	0.029	0.996	1.36	28.26	0.0011	0.997	1.359
40	35.1	33.30	0.074	0.985	1.33	36.25	0.0034	0.999	1.316
50	41.15	39.08	0.111	0.989	1.67	41.36	0.0056	0.999	1.673

**Table 3 ijerph-20-05750-t003:** Nonlinear parameters of pseudo-first-order and pseudo-second-order for the adsorption of CV dye in MB/CV binary system onto Cu-doped ZnO/Ext at pH 7.5 and at room temperature.

Nonlinear Pseudo-First-Order						Nonlinear Pseudo-Second-Order			
[MB] ppm	$q_{e,\;\;exp\;}$ (mg/g)	$q_{e,cal\;}$ (mg/g)	$k_{1\;}$ (min^−1^)	$R^{2}$	RSS	*q_e. (cal)_* (mg/g)	$k_{2\;}$ (min^−1^)	$R^{2}$	RSS
10	7.55	6.99	0.042	0.961	1.505	8.00	0.007	0.988	0.472
20	15.88	15.18	0.041	0.982	3.234	17.45	0.003	0.997	0.502
30	23.67	23.01	0.025	0.978	8.887	28.19	0.001	0.991	3.510
40	29.55	28.8	0.108	0.995	3.382	30.37	0.008	1.000	0.194
50	41.65	38.2	0.067	0.973	31.234	42.03	0.003	0.994	7.040

**Table 4 ijerph-20-05750-t004:** Nonlinear and linear parameters of Elovich model for the adsorption of MB dye in MB/CV binary system onto Cu-doped ZnO/Ext at pH 7.5 and at room temperature.

Nonlinear Elovich Model						Linear Elovich Model			
[MB] ppm	$q_{e,exp\;}$ (mg/g)	*β* (g/mg)	*α* (mg/g/min)	$R^{2}$	RSS	*β* (g/mg)	*α* (mg/g/min)	$R^{2}$	RSS
10	8.55	0.227	79.071	0.997	2.542	0.585	1.709	0.974	1.137
20	17.2	0.889	12.25	0.999	0.052	0.272	3.672	0.975	5.099
30	23.33	0.246	2.080	0.976	5.780	0.2080	4.807	0.962	13.647
40	35.1	0.152	1.558	0.988	5.112	0.139	7.211	0.971	22.971
50	41.15	0.296	40.571	0.999	0.679	0.117	8.560	0.929	82.477

**Table 5 ijerph-20-05750-t005:** Nonlinear and linear parameters of Elovich model for the adsorption of CV dye in MB/CV binary system onto Cu-doped ZnO/Ext at pH 7.5 and at room temperature.

Nonlinear Elovich Model						Linear Elovich Model			
[MB] ppm	$q_{e,exp\;}$ (mg/g)	*β* (g/mg)	*α* (mg/g/min)	$R^{2}$	RSS	*β* (g/mg)	α (mg/g/min)	$R^{2}$	RSS
10	8.55	1.25	0.666	0.998	0.063	0.686	1.460	0.992	0.075
20	17.2	2.49	0.300	0.997	0.490	0.309	3.241	0.990	0.431
30	23.33	1.22	0.1462	0.997	1.291	0.166	6.032	0.987	2.109
40	35.1	79.071	0.227	0.996	2.411	0.429	2.331	0.950	2.410
50	41.15	38.25	0.171	0.999	1.049	0.171	5.835	0.993	1.037

**Table 6 ijerph-20-05750-t006:** Nonlinear parameters and function errors of Langmuir and Freundlich model for the adsorption of MB dye in MB/CV binary system onto Cu-doped ZnO/Ext at pH 7.5 and at room temperature.

Dye in Mixture		Langmuir Constants			Freundlich Constants			
	$q_{m}$ (mg/g)	$K_{L}$ (L/mg)	$R^{2}$	RSS	$K_{F}$ (L/mg)	*n*	$R^{2}$	RSS
MB	46.61	0.077	0.982	22.38	21.42	0.352	0.954	35.88
CV	72.49	0.836	0.998	2.14	6.6	0.673	0.987	4.84

## Data Availability

Data are available from the corresponding author upon reasonable request.

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
