# Peer review of "Surface Functionalization of Bioactive Hybrid Adsorbents for Enhanced Adsorption of Organic Dyes"

_ijerph, 2023, doi:10.3390/ijerph20095750_

Round 1

Reviewer 1 Report

The manuscript "Surface Functionalization of Bioactive Hybrid Adsorbents for Enhanced Adsorption of Organic dyes" uses commercial ZnO and mango seed extract (MS-Ext) to synthesize a valuable adsorbent for the adsorption of organic dyes. In some ways, it is very observant of the relevant field, and I recommend publishing it. However, there are still some problems to be solved and improved:

1、The structure of the whole paper is very messy and there are many mistakes in the article. For example, ZnO/MS-Ex in line 139 should be changed to ZnO/MS-Ext; Line 123 should be deleted.

2、Figure 2(a) The FTIR spectra drawing is very messy. Meanwhile, it is suggested to add the FTIR spectra of Cu-doped ZnO/MS-Ext and discuss the changes of the spectrum before and after adsorption.

3、The novelty of the work needs to be established and emphasized.

4、How the adsorbent is regenerated and the service life should be also considered.

5、Please explain the following experimental phenomena in detail: first, Figure 9(a) In the binary adsorption system, the adsorption efficiency decreases when PH is neutral; second, Figure 9(c) the adsorption efficiency decreased with the increase of PH.

Reviewer 2 Report

Please correct all grammatical and syntax errors.
